# Current Understanding of *DDX41* Mutations in Myeloid Neoplasms

**DOI:** 10.3390/cancers15020344

**Published:** 2023-01-05

**Authors:** Kunhwa Kim, Faustine Ong, Koji Sasaki

**Affiliations:** Department of Leukemia, The University of Texas MD Anderson Cancer Center, Houston, TX 77030, USA

**Keywords:** *DDX41* mutations, myelodysplastic syndrome, acute myeloid leukemia, hereditary myeloid neoplasms

## Abstract

**Simple Summary:**

The DEAD-box RNA helicase 41, *DDX41*, is one of the most frequently identified mutations in myeloid neoplasms with germline predispositions, which represents 2% of the entire MDS/AML population. *DDX41* is located at 5q35.3, and its mutation has unique features of male predominance and long-term cytopenia before the development of myeloid neoplasms. So far, mechanism studies revealed that *DDX41* mutations, affected by both germline and somatic mutations, can be pathogenic by impairments in the normal function of genes involving RNA splicing and processing, ribosomal biogenesis, metabolism, cycle progression, and innate immunity. We are gaining a better understanding of disease from more studies coming out with larger cohorts. The survival impact of the mutation remains unclear, although recent larger studies suggest a better treatment response and survival in higher risk MDS/AML. Several studies showed a good response to lenalidomide in certain patients with MDS with *DDX41* mutations. Early identification of stem-cell transplant donors in the family for patients with *DDX41* mutations is crucial to avoid donor-derived leukemia from germline carriers. In this article, we reviewed the current understanding of DDX41 mutations in AML/MDS, including its pathogenesis and clinical characteristics, outcome, and treatment.

**Abstract:**

The DEAD-box RNA helicase 41 gene, *DDX41*, is frequently mutated in hereditary myeloid neoplasms, identified in 2% of entire patients with AML/MDS. The pathogenesis of *DDX41* mutation is related to the defect in the gene’s normal functions of RNA and innate immunity. About 80% of patients with germline *DDX41* mutations have somatic mutations in another allele, resulting in the biallelic *DDX41* mutation. Patients with the disease with *DDX41* mutations reportedly often present with the higher-grade disease, but there are conflicting reports about its impact on survival outcomes. Recent studies using larger cohorts reported a favorable outcome with a better response to standard therapies in patients with *DDX41* mutations to patients without *DDX41* mutations. For stem-cell transplantation, it is important for patients with *DDX41* germline mutations to identify family donors early to improve outcomes. Still, there is a gap in knowledge on whether germline DDX41 mutations and its pathology features can be targetable for treatment, and what constitutes an appropriate screening/surveillance strategy for identified carriers. This article reviews our current understanding of *DDX41* mutations in myeloid neoplasms in pathologic and clinical features and their clinical implications.

## 1. Introduction

The assessment of next-generation sequencing in the study of myelodysplastic syndromes (MDS) and acute myeloid leukemia (AML) is now guiding treatment decisions and predicting survival along with age and comorbidity considerations at the time of diagnosis [1,2,3,4,5,6,7,8,9,10,11,12,13,14,15,16,17,18,19]. The application of targeted therapies with *FLT3* inhibitors, *IDH* inhibitors, and venetoclax has further improved outcomes in patients with MDS and AML in newly diagnosed and relapsed settings [20,21,22,23,24,25,26,27,28,29,30,31].

In recent years, the accumulation of genetic data in myeloid neoplasms from next-generation sequencing has resulted in a rapid gain in understanding of hereditary myeloid neoplasms and identification of their related pathogenic germline mutations [32]. Patients with myeloid neoplasms with germline predisposition represent approximately 5–15% of all adult MDS and AML cases. The World Health Organization designated a new category of myeloid neoplasms with germline predisposition in 2016 [33].

The DEAD-box RNA helicase-1 gene (*DDX41*) is one of the most frequently identified mutations in myeloid neoplasms with a genetic predisposition. The *DDX41* gene is located at chromosome 5q35.3, and the gene can be affected by germline and somatic mutations [34]. Large cohort studies have shown that 1.5–3.8% of patients with myeloid neoplasms have *DDX41* mutations [35,36,37]. First identified in 2015 as predisposing mutations for myeloid neoplasms by a study of an index family with a strong history of MDS/AML, a significant proportion of hereditary myeloid neoplasms were associated with the *DDX41* mutations by affecting RNA biology and innate immunity [34]. Many studies have come out in recent years, which provide a better understanding of pathogenesis and clinical implications of *DDX41* mutations. In particular, as a common mutation in the association of hereditary myeloid neoplasms, understanding *DDX41* mutations can help establish the strategy for surveillance and management of germline mutation carriers, and screening and early identification for stem-cell transplant donors for the patients with MDS/AML with germline *DDX41* mutations.

This article reviews our current understanding of *DDX41* mutations in MDS/AML, focusing on pathogenic mechanisms, and pathologic and clinical data.

## 2. *DDX41* Mutations and Their Role in MDS/AML Pathogenesis

The pathology of *DDX41* mutations in myeloid neoplasms may be related to a disruption of the gene’s normal functions involving multiple functions in RNA biology [34]. *DDX41* encodes a member of the DEAD-box ATP-dependent RNA helicases, which are involved in pre-mRNA splicing, RNA processing, ribosome biogenesis, and small nucleolar RNA processing [34,38,39]. In addition, *DDX41* interacts with intracellular DNA in dendritic cells and macrophages, and activates innate immunity through the stimulator of interferon genes (STING)-interferon pathway [40,41]. More studies have come out in past years describing and identifying the pathomechanism of *DDX41* mutations.

Changes in *DDX41* expression have demonstrated variable effects in vitro and in vivo. For example, the knockdown of *DDX41* in K562 cells was shown to increase their proliferation in vitro and accelerate their tumor growth in a xenograft model [34]. Conversely, the knockout of *DDX41* in a mouse model impaired the differentiation and development of hematopoietic cells, particularly myeloid lineage cells, and decreased their proliferation capacity both in vivo and ex vivo [42].

A recent mouse model study demonstrated that monoallelic germline mutations of *DDX41* cause age-dependent myelodysplastic changes, whereas biallelic mutations resulting from the acquisition of somatic mutations on the other allele cause hematopoietic defects at a young age, likely by dysregulating small nucleolar RNA processing and ribosomal function [39]. *DDX41* mutations may also have an oncogenic effect by dysregulating the STING-interferon pathway, thereby impairing innate immunity [40,43].

Additionally, Mosler et al. proposed that an increased level of R-loop, RNA-DNA hybrids and displaced strand of DNA can be caused by dysregulation from loss of *DDX41*, as outlined in a recent study using quantitative mass spectrometry-based proteomics using human cells [44]. The study showed pathogenic variants of *DDX41* mutations resulted in the accumulation of double-strand breaks in human hematopoietic stem cells causing inflammatory responses. These enhanced inflammatory responses from R-loop accumulation and the dysregulation of hematopoietic stem cell production were also shown in a study using zebra fish by Weinreb et al. [45]. These studies propose the role of *DDX41* in genomic stability against R-loop generations.

*DDX41* mutations are clinically associated with an increased lifetime risk of myeloid neoplasms, including the early presentation of idiopathic cytopenia of undetermined significance (ICUS) [46]. Diseases reported to be associated with *DDX41* mutations include myeloproliferative neoplasms, chronic lymphocytic leukemia, chronic myeloid leukemia, multiple myeloma, and Hodgkin and non-Hodgkin lymphomas other than myeloid neoplasms [35,47,48,49]. *DDX41* mutations are also associated with blood disorders, including macrocytosis [47]. Some studies found *DDX41* mutations to be associated with an increased risk of autoimmune disorders or solid cancers, but these findings require further investigation [35,47]. The specific risk for hematologic malignancies including MDS/AML conveyed by *DDX41* mutations remains unclear.

## 3. Pathologic Features of Myeloid Neoplasms with *DDX41* Mutations

In common practice, next-generation sequencing panels at diagnosis of myeloid neoplasms or follow-up can detect *DDX41* mutations [47]. Comprehensive genomic testing with exome or genome sequencing can be used to identify variants that gene-targeted sequencing might miss [47]. The results of germline pathogenic variant testing can be confirmed by specimen without blood contamination obtained from skin fibroblasts [47].

In a recently published study using a multi-national cohort, *DDX41* germline mutations account for about 80% of patients with myeloid neoplasms with germline predisposition [37]. The study further showed that pathologic or likely pathologic germline mutations of DDX41 were identified 10 times more frequently in patients with myeloid neoplasms than the general population [37]. Germline *DDX41* mutations are passed from parent to child through autosomal dominant inheritance, but the penetrance of *DDX41* mutations remains unclear. Only about 30% of patients with germline *DDX41* mutations have a family history of hematologic malignancy, but a familial series study showed high penetrant patterns in some families [35,49]. Interestingly, the *DDX41*-mutant MDS/AML has a male predominance, with a male-to-female ratio of 3:1 [35,50]. The etiology of male predominance remains unknown.

Up to 80% of carriers of germline *DDX41* mutations who develop MDS/AML have additional somatic mutations on the other *DDX41* allele, which has been reported by multiple studies [34,49,50]. The pattern of biallelic mutations from secondary somatic mutations in *DDX41* mutations also can be seen in somatic mutations in the CCAAT enhancer binding protein alpha gene, *CEBPA,* in AML [34]. The role of additional somatic mutation and biallelic mutation as pathogenesis of *DDX41* mutations were described by Chlon et al. as mentioned above.

Notably, germline and somatic *DDX41* mutations tend to have different patterns and locations of mutations on the gene.

Most patients with germline mutations have pathogenic variants with either nonsense, frameshift, or splicing site mutations [49,51]. One of the most frequently identified germline mutations, p.D140fs, causes protein truncation [52]. Other germline mutations are associated with loss of function and derangements in splicing [49]. A study of 346 patients with germline *DDX41* mutations reported that MDS patients with truncating variants of *DDX41* germline mutations were observed to have shorter duration to AML transformation, about 2.5 times faster, compared to patients with non-truncating variants [37]. However, there was no difference in overall survival between the two groups [37]. The most frequent somatic mutation is the p.R525H missense mutation [36,51]. The mutation is located at the C-terminus of the helicase domain at the site of ATP interaction and hydrolyzation, which can cause deranged interactions with ATP in the helicase domain without directly changing the main domain [38]. One study showed that cord blood cells and leukemia cells harboring the p.R525H mutation had deranged pre-mRNA processing and ribosomal biogenesis from mutation-related change, resulting in impaired E2 factor activity and, thus, defective cell cycle progression [38].

In a next-generation sequencing study of patients with *DDX41*-mutant disease, most germline mutations (93%) were upstream of the helicase 2 domain or involved loss of the start codon (30%). In contrast, most somatic mutations (78%) were within the helicase 2 domain [36].

Ethnicity-based differences in the frequencies of *DDX41* mutations have been noted. Several studies from cohorts of Asian patients showed a distinct mutations profile. In a study of 28 Korean patients, the most common germline mutation was p.V152G, which was detected in 10 patients, followed by p.Y259C, p.A500fs, and p.E7* [46]. Similarly, in a Japanese population, p.A500fs and p.E7* were the most frequent germline mutations, and p.R525H was the most common somatic mutation [53]. In a study of Chinese patients, the most common mutations were c.935 + 4A>T and p.T360Ifs*33, which were detected in three patients each [54]. This ethnic difference in mutations was shown in a recent study by Li et al. with large data including 176 patients with germline mutations [50]. In the study, p.A500fs was only identified in Asian patients, and 92% of patients with Y295C (*n* = 12) were Asian patients. The majority, over 90%, of patients with p.M1I or p.D140fs were Caucasians. Interestingly, the study also showed a more frequent loss of function in germline mutations (85% vs. 51%) in Caucasians than in Asians.

Prior reports of mutation-related protein changes are summarized in Table 1. The study by Makishima et al., was published upon acceptance of our manuscript, and was not included in the table [37].

Bone marrow examination often shows hypocellularity and erythroid dysplasia [46,52]. 70–80% of patients with germline *DDX41* mutations have a normal karyotype [34,35,36]. In addition, 3–30% of patients have concomitant *TP53* mutations [35,36,56]. A systematic review study of pooled studies of *DDX41* mutations showed 47% of patients have concomitant other somatic mutations, most frequently *ASXL1*(26%), *TP53* (23%), followed by *TET2*, *EZH2*, *SRSF2,* and *DNMT3A* [48]. Interestingly, the recent study by Makishima et al. including 346 patients with pathogenic or likely pathogenic germline *DDX41* mutations showed that co-mutations, including TP53, did not affect outcomes [37].

## 4. Clinical Presentation and Outcomes

The age at diagnosis of MDS/AML patients with germline *DDX41* mutations has been reported within a wide range from 57 to 70 years, but most studies reported that it is comparable to patients with general MDS/AML [34,35,36,50,58].

Some studies showed that MDS/AML patients with germline *DDX41* mutations often have a higher-grade disease than those with wildtype *DDX41* [34,36,51]. However, the survival effect of germline *DDX41* mutations was reported to vary.

Several studies have assessed outcomes in MDS/AML patients with *DDX41* mutations. Polprasert et al. showed that MDS/AML patients with *DDX41* mutations had worse survival than patients with wildtype *DDX41* in a large cohort of MDS and secondary AML patients of 1034 patients (hazard ratio, 3.5; *p* < 0.0001) [34]. In a smaller study that included 28 patients with ICUS, MDS, or AML having *DDX41* germline mutations, Choi et al. found no difference in survival by *DDX41* mutation status [46].

Sebert et al. showed that the median survival duration of patients with germline *DDX41* mutations (5.2 years) was longer than that of patients with wildtype *DDX41* (2.7 years) in a propensity score-matched study with 18 matched patients with *DDX41* mutations, but this difference was not statistically significant. They also found that patients with germline mutations had a good overall response to intensive chemotherapy (response rate at 100%, *n* = 9) and hypomethylating agent (response rate at 73%, *n* = 11), and a median response duration of 2.5 years [35].

Li et al. showed that 81 patients with *DDX41* germline causal variants had superior overall survival compared to age-matched or general MDS or AML cases with wild-type *DDX41* or a variant of unknown significance (median OS: not reached) [50].

A recent study by Duployez et al. showed prolonged survival in *DDX41* germline mutant AML patients, with a good response to intensive chemotherapy in intermediate or high-risk patients with AML, in a large cohort study using five prospective clinical trials of 191 newly diagnosed *DDX41* mutant AML patients [57]. The median overall survival of AML patients with *DDX41* germline mutations of all variants was 28.1 months (Interquartile range, IQR, at 10.7–82.7 months) when censored at stem-cell transplant, and median relapse-free survival at 18.7 months (IQR 8.3–80.9 months).

These conflicting results are likely due to different study populations, heterogeneity of MDS/AML clinical course and treatments, and a limited number of study patients from a low prevalence of mutations. While Sebert et al. studied patients with germline *DDX41* mutations and analyzed the survival of only those patients who were treated for high-risk MDS and AML, Polprasert et al. included all patients with either germline or somatic mutations in the survival analyses. Choi et al. included patients with either germline or somatic mutations, but analyzed survival by diagnosis. Li et al. studied patients with germline pathogenic variants. Duployez et al. studied AML patients only.

As the prevalence of *DDX41*-associated MDS/AML is relatively low, further accumulation of clinical data may help clarify the prognostic role of *DDX41* mutations.

## 5. Treatment—General Approaches

So far, there are no randomized studies focusing on patients with *DDX41* mutations for specific therapeutic approaches. The treatment approaches of *DDX41*-mutant myeloid neoplasms generally follow standard care for general MDS/AML treatments. Two recent studies reported treatment responses in *DDX41* mutant patients. Li et al. reported a higher response rate at 78%, and superior survival (not reached) in a matched case-control study including 28 patients with AML having *DDX41* germline mutations. In the study, most (about 80%) patients were treated with hypomethylating agents with or without venetoclax [59]. Duployez et al., as mentioned above, also reported a better response to intensive chemotherapy in patients with intermediate or adverse *DDX41* germline-mutant patients in AML.

Interestingly, a few studies showed a good response to lenalidomide in MDS patients with *DDX41* mutations [34]. Lenalidomide is a well-established standard therapy for the low-risk MDS with a 5q deletion [60]. Lenalidomide has demonstrated its efficacy in MDS/AML patients with *DDX41* mutations, even in the absence of a full 5q deletion [34]. Only 26% of patients with MDS with 5q deletion have the deletion at 5q35, where *DDX41* is located at [34].

In a study of 111 patients treated with lenalidomide, Polprasert et al. reported that the response rate among patients with *DDX41* mutations (100% (8/8 patients)) was significantly higher than that among patients with wildtype *DDX41* (53% (55/103 patients); *p* = 0.01). In another prediction model study using 137 patients with MDS or other myeloid neoplasms, the mutation of any DEAD-box RNA helicase gene, including *DDX41,* was associated with higher response to lenalidomide (odds ratio 3.4, *p* = 0.04). Among seven patients with heterozygous *DDX41* mutations or deletions, the lenalidomide response rate was 57%. For the three patients who had *DDX41* mutations only, the response rate was 100% [61].

In addition, there is a case report about single-agent lenalidomide with the successful treatment of one patient with high-risk MDS (i.e., MDS with excess blasts type 2) [62]. The patient, who had a germline mutation (p.D140fs) and a low burden of *DDX41* p.R525H mutation, had a good response and had improved blood counts and rare dysplasia after 4 months of lenalidomide monotherapy. The blast percentage had decreased from 16% before treatment to 3% [62].

## 6. Treatment—Special Considerations: SCT

SCT is an essential part of treatment in hereditary myeloid neoplasms, especially in fit, younger patients. Donor cell leukemia is a particular concern in *DDX41* germline mutant myeloid neoplasms with its severity and fatality.

Berger et al. were the first to report a case of a *DDX41* mutation giving rise to donor cell leukemia. They described a patient who received an allogeneic SCT from an HLA-matched related donor carrying a germline *DDX41* mutation (c.3G>A; p.(MET1?)) and other somatic mutations [63]. The patient’s family had a strong history of leukemia; the patient’s father died of leukemia, and one of the patient’s two brothers (not the donor) developed AML. The donor remained free of disease, but the patient developed high-risk MDS that might have been associated with stress from post-transplant changes [63].

Kobayashi et al. similarly reported a case of donor cell leukemia in a patient who received stem cells from a donor with a germline *DDX41* mutation (p.F498fs). There was also a significant increase in the variant allele frequency (7.9% vs 0.4%) of a somatic *DDX41* mutation (p.R525H) in a carrier after transplant [64]. Larger studies have since investigated the effects of donor clonal hematopoiesis in SCT. In a recent study of 1727 patients who received stem cells from donors with clonal hematopoiesis, only eight cases of donor cell leukemia were identified; in two of these cases, germline *DDX41* mutations were present [65].

Therefore, the early identification of the related donors’ germline *DDX41* mutation carrier status is one strategy to avoid SCT delay in MDS/AML patients with germline *DDX41* mutations. In a report on the experience of a single tertiary cancer center, Bannon et al. reported that such identification of optimal related donors was effective [51].

## 7. Family Screening and Surveillance

There are no specific guidelines for family screening, or the surveillance of germline carriers identified from family screening. Family screening might be considered only for patients with germline mutations because somatic *DDX41* mutations are not heritable, and *DDX41* mutations are rare among the general population [52]. However, the risk of malignancy and penetrance of disease in germline carriers remains unclear, and methods for detecting or preventing myeloid neoplasms and other cancers in these individuals have not been established [66].

As a first step, surveillance methods and the timing for the family members identified as germline *DDX41* need to be established. Lewinsohn et al. reported that germline *DDX41* mutation carriers had cytopenia and other complete blood count (CBC) abnormalities at the time of MDS/AML diagnosis, but mostly lacked these abnormalities before diagnosis [49]. Further studies showed earlier CBC changes in carriers; Polprasert et al. reported macrocytosis and monocytosis in carriers, and Bannon et al. and Sebert et al. reported that 50–60% of carriers who were later diagnosed with hematologic malignancies had antecedent cytopenia [34,35,51]. In a study by Choi et al., five of 28 patients with germline *DDX41* mutations had ICUS [46]. Therefore, CBCs can be helpful in the screening or surveillance of carriers identified from family screening, but specific recommendations are not yet available. Makishima et al., reported an increasing risk of developing myeloid neoplasms by age in germline carriers [37]. The risk for myeloid neoplasms in patients with three major pathogenic variants under the age of 40 were negligible, but it increases rapidly after the age of 40 [37]. Age 40 could be a cut-off for a surveillance in germline carriers based on study results, although further accumulation of data would be helpful.

Several papers have described the experiences of patients with germline *DDX41* mutations in genetics clinics. One review, which detailed multiple cases from MD Anderson’s genetics clinic, described the experience of a patient who had a biallelic *DDX41* mutation with a hotspot germline *DDX41* mutation and the screening of the patient’s adult children [66]. Bannon et al. described how *DDX41* mutation carriers were offered education and follow-up; all unaffected family members, including potential stem cell transplant (SCT) donors, were offered genetic testing and counseling [51]. This enabled the prompt transition to stem-cell donor identification within 15 days after family screening [51].

Other considerations for the screening and surveillance of patients and their families should include the early detection of secondary hematologic malignancies, both myeloid and lymphoid malignancies, including follicular lymphoma and Hodgkin’s lymphoma. It should also include autoimmune diseases in MDS/AML patients who have had disease remission, and the early detection of primary hematologic malignancies or autoimmune diseases in germline mutation carriers.

## 8. Conclusions

We are rapidly gaining knowledge about hereditary MDS/AML through recent clinical and translational research endeavors, particularly of *DDX41*-associated hereditary myeloid neoplasms. MDS/AML with *DDX41* mutations has become a study of interest with its unique features of pathophysiology, genetic and clinical characteristics. There is the potential therapeutic implication of an expected good treatment response to standard therapies, and the need for early donor identifications for SCT.

Despite the rapid advancements in this field, there still is a gap in knowledge that future studies need to address for patients with *DDX41*-associated myeloid neoplasms. Further understanding of the clinical impact of the *DDX41* mutations, including learning further prognostic or clinical information of *DDX41* mutant AML/MDS patients, and studying whether *DDX41* mutations can be targetable for therapeutic or preventive applications would be essential. Additionally, establishing standardized approaches for cancer and other medical surveillance/screening in family members (i.e., germline-carriers) is essential.

## Figures and Tables

**Table 1 cancers-15-00344-t001:** Summary of *DDX41* studies and identified germline mutations.

Study Characteristics (Author, Year, Cohort, Database)	Total Number of Data	Nucleotide Change	Amino Acid Change	Number of Patients or Families (% of Studied Patients or Families)	No. of Patients with Concomitant Somatic *DDX41* Mutations	No. of Patients with Hematologic Malignancies
Polprasert et al., 2015 [34]; a cohort of MDS/secondary AML, multicenter (US/Germany), and TCGA database	27/1034 patients and 7 index families (19 patients with germline mutation)	c.419insGATG (c.415_418dupGATG)	p.D140fs	14 (74%)	5/14	8 AML 6 MDS/CMML
		c.T1187C	p.I396T	2 (10%)	2/2	2 MDS
		c.156_157insA	p.Q52fs	1 (5%)	1/1	1 AML
		c.G465A	p.M155I	1 (5%)	0/1	1 MDS
		Not mentioned	p.F183I	1 (5%)	1/1	1 MDS
Lewinsohn et al., 2016 [49]; a cohort of families with suspected inherited hematologic malignancies, multicenter (Australian/US) familial hematologic malignancies registry	9/289 families	c.415_418dupGATG	p.D140Gfs	3 families	Not reported	3 AMLs
		c.3G>A	p.M1I	2 families	Not reported	3 AML (1 with NHL involvement), 1 MDS, 1 CML
		c.435-2_435-1delAGinsCA	(predicted to produce p.W146Hfsand p.S145Rfs)	1 family	Not reported	1 MDS
		c.490C>T	p.R164W	1 family	Not reported	3 NHL
		c.1574G>A	p.R525H (suspected germline)	1 family	Not reported	2 MDS, 1 AML
		c.1589G>A	p.G530D	1 family	Not reported	3 AML
Cardoso et al., 2016 [55]; a cohort of families with at least two cases of bone marrow failure and at least one of whom having MDS or AML (no detailed description of the study cohort)	4/78 families	c.3G>A	p.M1I	1 family	Not reported	2 MDS
		c.155dupA	p. R53Afs	1 family	Not reported	3 MDS, 1 carrier (1 CML family history with unchecked mutation)
		c.719delTinsCG	p.I240Tfs	1 family	Not reported	1 AML (1 AML family history with unchecked mutation)
		c.1586-1587delCA	p.T529Rfs	1 family	Not reported	1 MDS, 1 carrier (1 AML family history with unchecked mutation)
Sebert et al., 2019 [35]; a cohort of families with a family history of MDS, AML, AA, single-center (France) data	43/1385 patients (33 patients with causal germline variants)	c.G517A	p.G173R	6 (18%)	6/6	3 MDS, 1 AML, 2 AA
		c.G3A	p.M1I	3 (9%)	3/3	2 AML, 1 MDS
		c.992_994del	p.K331del	3 (9%)	2/3	1 MDS/MPN, 1 MDS, 1 AML
		c.C121T	p.Q41*	2 (6%)	1/2	2 MDS
		c.418_419insGATG	p.D140fs	2 (6%)	1/2	1 AML, 1 MDS/MPN
		c.C1015T	p.R339C	2 (6%)	1/2	1 AA, 1 MDS
		c.A1C	p.M1L	1 (3%)	1/1	1 MDS
		c.69delC	p.S23fs	1 (3%)	1/1	1 MDS
		c.A316T	p.K106*	1 (3%)	0/1	1 CMML
		c.342_346del	p.E114fs	1 (3%)	1/1	1 MDS
		c.542+2A>G	-	1 (3%)	0/1	1 MDS
		c.644+1G>A	-	1 (3%)	1/1	1 MDS
		c.T649C	p.S217P	1 (3%)	1/1	1 MDS
		c.A734G	p.E245G	1 (3%)	1/1	1 MDS
		c.799-2T>A	-	1 (3%)	1/1	1 AML
		c.945delC	p.H315fs	1 (3%)	1/1	1 AML
		c.A1031G	p.D344G	1 (3%)	1/1	neutropenia only
		c.1088_1090del	p.S363del	1 (3%)	1/1	1 AML
		c.C1108T	p.Q370*	1 (3%)	1/1	1 AML
		c.1298dupC	p.P433fs	1 (3%)	0/1	1 AML
		c.1791_1792del	p.K597fs	1 (3%)	1/1	1 AML
Quesada et al., 2019 [36]; a cohort of known/suspected myeloid neoplasms, single-center (US) data	34/1002 patients (32 patients with germline mutations)	c.3G>A	p.M1I	9 (28%)	9/9	1 AML, 4 MDS->AML, 4 MDS
		c.415_418dupGATG	p.D140Gfs	4 (13%)	4/4	2 AML, 1 MDS->AML, 1 MDS
		c.121C>T	p.Q41*	2 (6%)	2/2	2 MDS
		c.25A>G	p.K9E	1 (3%)	0/1	1 MDS/CMML->AML
		c.38C>T	p.T13I	1 (3%)	0/1	1 Post PV-MF
		c.59G>A	p.G20E	1 (3%)	0/1	1 MDS
		c.62_63del	p.S21Tfs	1 (3%)	1/1	1 AML
		c.142C>T	p.Q48*	1 (3%)	1/1	1 MDS->AML
		c.298+2_298+4delTGG	Splice	1 (3%)	1/1	1 MDS->AML
		c.475C>T	p.R159*	1 (3%)	1/1	1 MDS->AML
		c.476G>A	p.R159Q	1 (3%)	0/1	MPN
		c.572-1G>A	Splice	1 (3%)	1/1	1 AML
		c.608A>G	p.H203R	1 (3%)	0/1	1 MDS->AML
		c.649T>C	p.S217P	1 (3%)	1/1	1 AML
		c.821A>G	p.H274R	1 (3%)	0/1	1 MPN
		c.1046T>A	p.M349K	1 (3%)	1/1	1 suspected MDS
		c.1105C>T	p.R369*	1 (3%)	1/1	1 MDS->AML
		c.1105C>G	p.R369G	1 (3%)	1/1	1 MDS
		c.1771C>T	p.R591W	1 (3%)	0/1	1 MDS->AML
		c.1766G>A	p.G589D	1 (3%)	0/1	1 CMML
Choi et al., 2021 [46]; a cohort of patients with ICUS/MDS/AML, single-center (Korea) data	39/457 patients (34 patients with germline mutations)	c.455T>G	p.V152G	10 (29%)	10/10	2 ICUS, 8 MDS
		c.776A>G	p.Y259C	9 (26%)	8/9	2 ICUS, 7 MDS
		c.1496dupC	p.A500fs	6 (18%)	6/6	1 ICUS, 2 MDS, 3 AML
		c.19G>T	p.E7*	3 (9%)	2/3	2 MDS, 1 AML
			p.D139G	2 (6%)	0/2	1 MDS, 1 AML
			p.E3K	1 (3%)	0/1	1 AML
			p.Y33C	1 (3%)	0/1	1 AML
			p.K187R	1 (3%)	0/1	1 AML
		c.983T>G	p.L328R	1 (3%)	1/1	1 MDS
Bannon et al., 2021 [51]; a cohort of patients who were referred to genetic counseling and testing for hematologic malignancies with *DDX41* mutations, single-center (US) data	33 (38 referred)/90 *DDX41* germline mutations (out of 5801 heme malignancies patients)	c.415_418dupGATG	p.D140fs	10 (30%)	7/10	5 AML, 4 MDS (1 carrier)
		c.3A>G	p.M1I	8 (24%)	2/8	4 AML, 1 MDS->AML, 2 MDS, 1 CLL
		c.121C>T	p.Q41*	3 (9%)	2/3	2 AML, 1 MDS
		c.337del	p.E113fs	1 (3%)	1/1	1 MDS->AML
		c.434+1G>A	-	1 (3%)	0/1	1 MDS
		c.475C>T	p.R159*	1 (3%)	0/1	1 MDS/MPN
		c.547T>G	p.F183V (VUS)	1 (3%)	1/1	1 MDS
		c.572-1G>A	-	1 (3%)	1/1	1 AML
		c.653G>A	p.G218D (VUS)	1 (3%)	0/1	1 MDS->AML
		c.847del	p.L283fs	1 (3%)	0/1	1 MDS
		c.946_947del	p.M316fs	1 (3%)	1/1	1 MDS
		c.1004dupT	p.D336fs	1 (3%)	1/1	1 MDS
		c.1105C>G	p.R369G (VUS)	1 (3%)	1/1	1 MDS
		c.1187T>C	p.I396T	1 (3%)	1/1	1 MDS
		c.1273_1276dupCTCG	p.E426fs	1 (3%)	1/1	1 AML
Qu et al., 2021 [54]; a cohort of myeloid neoplasms, single-center (China) data	47/1529 patients (25 patients with germline mutation)	c.1077_1078dupTA	p.T360Ifs	3 (12%)	3/3	3 MDS
		c.935+4A>T	-	3 (12%)	2/3	1 AML, 1 MDS, 1 Post-ET MF
		c. C1105T	p.R369*	2 (8%)	2/2	1 AML, 1 MDS
		c. T455G	p.V152G	2 (8%)	2/2	2 MDS
		c.647dupT	p.S217Ifs	2 (8%)	1/2	1 MDS, 1 Post-ET MF
		c. C931T	p.R311*	2 (8%)	2/2	2 MDS
		c. G3T	p.M1I	1 (4%)	1/1	1 MDS
		c. G391T	p.E131*	1 (4%)	1/1	1 MDS
		c. G553T	p.E185*	1 (4%)	1/1	1 AML
		c.572-1G>C	-	1 (4%)	1/1	1 MDS
		c. C773T	p.P258L	1 (4%)	1/1	1 MDS
		c.865delT	p.S289Hfs	1 (4%)	0/1	1 MDS
		c.1213_1216del	p.S405Wfs	1 (4%)	1/1	1 MDS
		c.1296_1298dup	p.P434dup	1 (4%)	1/1	1 MDS
		c. G1531T	p.E511*	1 (4%)	1/1	1 MDS
		c. A776G	p.Y259C	1 (4%)	1/1	1 MDS
		c.T983G	p.L328R	1 (4%)	1/1	1 MDS
Alkhateeb et al., 2022 [56]; a cohort of myeloid neoplasm patients, single-center (US) data	33/4524 patients (likely 25 germline patients)	c.3G>A	p.M1I	10 (40%)	1/10	4 AML, 5 MDS (1 carrier)
		c.415_418dup	p.D140Gfs	5 (20%)	0/5	4 MDS, 1 AML
		c.1589G>A	p.G530D	2 (8%)	0/2	1 MDS, 1 AML
		c.121C>T	p.Q41*	1 (4%)	0/1	1 AML
		c.305_306del	p.K102Rfs	1 (4%)	1/1	1 AML
		c.337del	p.E113Kfs	1 (4%)	0/1	1 MPN
		c.434+1G>A	-	1 (4%)	0/1	1 MDS
		c.776A>G	p.Y259C	1 (4%)	1/1	1 MDS
		c.931C>T	p.R311*	1 (4%)	1/1	1 AML
		c.946_947del	p.M316D	1 (4%)	0/1	1 MDS
		c.1102C>T	p.Q368*	1 (4%)	0/1	1 MDS
Li et al., 2022 [50]; a cohort of patients with hematologic malignancies, multi-center (US) data **	176/9821 patients, 116 patients with causal variants	c.3G>A	p.M1I	42 (36%)	35/42	24 AML, 13 MDS, 5 CCUS
		c.415_418dup	p.D140fs	23 (20%)	15/23	18 AML, 2 MDS, 1 MPN, 2 CCUS
		c.475C>T	p.R159*	3 (3%)	3/3	1 AML, 1 MDS, 1 CCUS
		c.931C>T	p.R311*	3 (3%)	3/3	1 AML, 2 MDS
		c.946_947del	p.M316fs	3 (3%)	2/3	1 AML, 1 MPN, 1 CCUS
		c.992_994del	p.K331del	3 (3%)	2/3	1 AML, 2 MDS
		c.1105C>T	p.R369*	3 (3%)	3/3	2 AML, 1 CCUS
		c.121C>T	p.Q41*	2 (2%)	1/2	2 AML
		c.773C>T	p.P258L	2 (2%)	2/2	1 AML, 1 CCUS
		c.1046T>A	p.M349K	2 (2%)	0/2	2 AML
		c.1105C>G	p.R369G	2 (2%)	2/2	1 AML, 1 MDS
		c.130C>T	p.Q44*	1 (1%)	1/1	1 MDS
		c.323del	p.K108fs	1 (1%)	1/1	1 AML
		c.430del	p.T144fs	1 (1%)	1/1	1 CCUS
		c.566C>T	p.P189L	1 (1%)	1/1	1 MDS
		c.645-1G>T	-	1 (1%)	1/1	1 AML
		c.646C>G	p.L216V	1 (1%)	1/1	1 AML
		c.649T>C	p.S217P	1 (1%)	1/1	1 CCUS
		c.653G>A	p.G218D	1 (1%)	1/1	1 AML
		c.668dup	p.I224fs	1 (1%)	1/1	1 AML
		c.710T>G	p.L237W	1 (1%)	1/1	1 MDS
		c.776A>G	p.Y259C	1 (1%)	1/1	1 CCUS
		c.847del	p.L283fs	1 (1%)	1/1	1 AML
		c.916C>T	p.Q306*	1 (1%)	0/1	1 MPN
		c.967C>T	p.R323C	1 (1%)	1/1	1 AML
		c.1015C>T	p.R339C	1 (1%)	1/1	1 MDS
		c.1016G>A	p.R339H	1 (1%)	0/1	1 MDS
		c.1016G>T	p.R339L	1 (1%)	1/1	1 CCUS
		c.1018T>A	p.Y340N	1 (1%)	1/1	1 MDS
		c.1108C>T	p.Q370*	1 (1%)	1/1	1 AML
		c.1141A>T	p.K381*	1 (1%)	1/1	1 MPN
		c.1354del	p.L452fs	1 (1%)	1/1	1 CCUS
		c.1394del	p.G465fs	1 (1%)	1/1	1 AML
		c.1399G>T	p.D467Y	1 (1%)	1/1	1 AML
		c.1496dup	p.A500fs	1 (1%)	1/1	1 MDS
		c.1504C>T	p.Q502*	1 (1%)	1/1	MDS
		c.1574G>A	p.R525H	1 (1%)	0/1	1 AML
		c.1586_ 1587delCA	p.T529fs	1 (1%)	0/1	1 CCUS
		c.1628C>G	p.S543*	1 (1%)	1/1	1 CCUS
Duployez et al, 2022 [57]; a cohort of 5 prospective trials and additional diagnostic samples, multi-center (France) data	191 AML patients with germline mutations	c.415_418dup	p.D140fs	32 (17%)	27/32	All AML patients
		c.3G>A	p.M1?	19 (10%)	15/19	
		c.517G>A	p.G173R	10 (5%)	8/10	
		c.847del	p.L283fs	9 (5%)	7/9	
		c.1088_1090del	p.S363del	9 (%)	8/9	
		c.138+1G>C		6	5/6	
		c.653G>A	p.G218D	5	5/5	
		c.992_994del	p.K331del	5	4/5	
		c.268C>T	p.Q90*	4	4/4	
		c.305_306del	p.K102fs	4	4/4	
		c.804del	p.E268fs	4	4/4	
		c.55G>T	p.G19*	3	2/3	
		c.121C>T	p.Q41*	3	3/3	
		c.936-1G>A	-	3	2/3	
		c.1212_1226delinsAG	p.S405fs	3	3/3	
		c.1334_1336del	p.V445del	3	2/3	
		c.1496del	p.P499fs	3	3/3	
		c.1A>C	p.M1?	2	2/2	
		c.316A>T	p.K106*	2	2/2	
		c.571G>A	p.A191T	2	2/2	
		c.656G>A	p.R219H	2	0/2	
		c.935+2T>C	-	2	1/2	
		c.945del	p.H315fs	2	1/2	
		c.1031A>G	p.D344G	2	2/2	
		c.1098+1G>A	-	2	2/2	
		c.1105C>T	p.R369*	2	1/2	
		c.1298del	p.P433fs	2	2/2	
		c.1504C>T	p.Q502*	2	1/2	
		c.1585dup	p.T529fs	2	1/2	
		c.2T>C	p.M1?	1	1/1	
		c.69del	p.S23fs	1	1/1	
		c.130C>T	p.Q44*	1	0/1	
		c.142C>T	p.Q48*	1	1/1	
		c.156_157delinsTT	p.Q52_R53delinsH	1	1/1	
		c.325C>T	p.Q109*	1	1/1	
		c.342_346del	p.E114fs	1	1/1	
		c.364G>T	p.E122*	1	1/1	
		c.373+1G>A	-	1	0/1	
		c.434+1G>T	-	1	1/1	
		c.436T>C	p.W146R	1	1/1	
		c.475C>T	p.R159*	1	1/1	
		c.622C>G	p.Q208E	1	1/1	
		c.643A>C	p.I215L	1	0/1	
		c.644T>C	p.I215T	1	0/1	
		c.649T>C	p.S217P	1	1/1	
		c.668G>A	p.G223D	1	1/1	
		c.758C>G	p.S252*	1	1/1	
		c.791G>A	p.C264Y	1	1/1	
		c.799-2A>T	-	1	1/1	
		c.805dup	p.L269fs	1	0/1	
		c.931C>T	p.R311*	1	0/1	
		c.958A>T	p.T320S	1	0/1	
		c.959C>T	p.T320I	1	1/1	
		c.967C>A	p.R323S	1	1/1	
		c.967C>T	p.R323C	1	0/1	
		c.968G>A	p.R323H	1	1/1	
		c.998T>A	p.V333D	1	1/1	
		c.1033G>A	p.E345K	1	1/1	
		c.1037C>A	p.A346D	1	1/1	
		c.1105C>G	p.R369G	1	1/1	
		c.1108C>T	p.Q370*	1	1/1	
		c.1118_1127del	p.L373fs	1	1/1	
		c.1212_1224del	p.L406fs	1	1/1	
		c.1252G>T	p.E418*	1	1/1	
		c.1298dup	p.P433fs	1	1/1	
		c.1463C>A	p.A488D	1	1/1	
		c.1514T>A	p.I505N	1	1/1	
		c.1615del	p.A539fs	1	1/1	
		c.1628C>G	p.S543*	1	1/1	
		c.1732+1del	-	1	1/1	
		c.1791_1792del	p.K597fs	1	1/1	

Abbreviation: AML: acute myeloid leukemia, ICUS: idiopathic cytopenia of undetermined significance, CML: chronic myeloid leukemia, CMML: chronic myelomonocytic leukemia, MDS: myeloid dysplastic syndrome, MPN: myeloproliferative neoplasm, NHL: non-Hodgkin’s lymphoma, post-PV MF: post-polycythemia vera myelofibrosis, post-ET MF: post-essential thrombocythemia myelofibrosis, VUS: variant of uncertain significance. ** Only included causal variants in the table (excluded VUS).

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
