# Peer review of "Current Understanding of DDX41 Mutations in Myeloid Neoplasms"

_cancers, 2023, doi:10.3390/cancers15020344_

Round 1

Reviewer 1 Report

This article highlight the current understanding of DEAD-box RNA helicase-1 gene (DDX41) mutations in MDS/AML. DDX41 is one of the  well-studied genes in hereditary myeloid neoplasms. Several studies have demonstrated that ~2% of patients with myeloid neoplasms have this specific mutations. Disruption of DDX41 is connected to cellular functions such as RNA biology and immune activation via STING interferon pathway. There are multiple reports citing the involvement of DDX41 mutations with blood disorders such as myeloproliferative neoplasms, chronic lymphocytic leukemia, chronic myeloid leukemia, multiple myeloma, and Hodgkin and non-Hodgkin lymphomas other than myeloid neoplasms.

The authors have nicely summarized many of the DDX41 studies and corresponding germline mutations identified. This is very informative.
Language of communication is professional and flawless.

I got curious with the existing treatment options highlighting the use of lenalidomide. However, the mechanism of action was not clear. How or why lenalidomide was better in hematological patients with DDX41 mutation or in patients with MDS patients with 5q deletion. Kindly add that information if available or atleast speculate them.

If possible, add a scheme for working mechanism, structural information of lenalidomide and alternate treatment options or add some representative schemes, graphs (with DDX41 stats) to make this overall manuscript exciting for larger audience.

This paper is relevant and interesting, I would recommend this paper for publication with minor changes. Congratulations to the authors.

Author Response

Thank you very much for your kind response. Your review encouraged us to further improve manuscript. Below is the updated part from your suggestions.

  1. Although lenalidomide was reported to have good therapeutic efficacy in a few reports, some reviewers pointed out that the studies included only a small number of patients, and there are no mechanisms known (limited to speculations by authors of small studies). We accepted the suggestions and updated the treatment options part into a more generalized understanding of treatment approaches to standard therapies, including lenalidomide. We re-wrote the part in italic font.

So far, there is no randomized studies focusing on patients with DDX41 mutations for specific therapeutic approaches. The treatment approaches of DDX41-mutant myeloid neoplasms generally follow standard care for general MDS/AML treatments. 2 recent studies reported treatment responses in DDX41 mutant patients. Li et al. reported higher response rate at 78% and superior survival (not reached) in matched case-control study including 28 patients with AML having DDX41 germline mutations. In the study, most (about 80%) of patients were treated with hypomethylating agents with or without venetoclax.59 Duployez et al., as mentioned above, also reported better response to intensive chemotherapy in patients with intermediate or adverse DDX41 germline-mutant patients in AML.

Interestingly, a few studies showed good response to Lenalidomide in MDS patients with DDX41 mutations36. Lenalidomide is a well-established standard therapy for the low-risk MDS with a 5q deletion60 Lenalidomide has demonstrated its efficacy in MDS/AML patients with DDX41 mutations, even in the absence of a full 5q deletion 36. Only 26% of patients with MDS with 5q deletion have the deletion at 5q35, where DDX41 is located at 36.

  1. We took more time to update our table to more accurately reflect recent studies. We hope these changes help readers understand so far studied mutations. We sincerely appreciate your input.

Again, thank you very much for taking the time to kindly review our paper. It was very helpful for our team.

Sincerely,

Kunhwa Kim, MD MPH

Koji Sasaki, MD PhD

Reviewer 2 Report

Kim et al. write a review on “DDX41 mutations in myeloid neoplasms”.

Actually, this is a review on a hereditary hematological malignancy, and this should be stated in the title with a word like “germline”.

I have major points to raise:

·      Two recent major publication in Blood on DDX41 in myeloid malignancies are not included in the review and should be integrated:

      - Duployez N, Largeaud L, Duchmann M, Kim R, Rieunier J, Lambert J, Bidet A, Larcher L, Lemoine J, Delhommeau F, Hirsch P, Fenwarth L, Kosmider O, Decroocq J, Bouvier A, Le Bris Y, Ochmann M, Santagostino A, Adès L, Fenaux P, Thomas X, Micol JB, Gardin C, Itzykson R, Soulier J, Clappier E, Recher C, Preudhomme C, Pigneux A, Dombret H, Delabesse E, Sébert M. Prognostic impact of DDX41 germline mutations in intensively treated acute myeloid leukemia patients: an ALFA-FILO study. Blood. 2022 Aug 18;140(7):756-768. doi: 10.1182/blood.2021015328. PMID: 35443031.

      - Li P, Brown S, Williams M, White T, Xie W, Cui W, Peker D, Lei L, Kunder CA, Wang HY, Murray SS, Vagher J, Kovacsovics T, Patel JL. The genetic landscape of germline DDX41 variants predisposing to myeloid neoplasms. Blood. 2022 Aug 18;140(7):716-755. doi: 10.1182/blood.2021015135. PMID: 35671390.

·      Regarding age at onset, authors stated in the simple summary “younger age at diagnosis than the general AML/MDS population” and in the text “For MDS/AML patients with DDX41 mutations, the median age at diagnosis is 60 years (range, 57–70 years), which is slightly lower than that for patients with general MDS/AML”. This is not what is reported in the cited studies. Sébert et al. (ref 33) reported a median age of 69 years, Quesada et al. (ref 34) 70 years and Polprasert et al. (ref 35) 64 years. This was confirmed in the two recent studies (see major point 1). In my knowledge, MDS/AML patients with germline DDX41 mutations are diagnosed at similar age than the sporadic patients DDX41 while type.

·      Regarding penetrance, the overall disease penetrance is currently not known as stated in recent review in the field: Klco et al, Nat review cancer (PMID: 33328584). We could speculate a low one, but no relevant study demonstrated it. In this way, authors should not claim that the penetrance is low.

·      Regarding biological functions of DDX41 protein. The authors resumed in the abstract and the simple summary as “RNA metabolisms and innate immunity”. In my opinion, the overall function of this protein in the context of haematopoietic cells and how these mutations lead to haematopoietic neoplasms is also related to RNA splicing and processing, ribosomal biogenesis and cycle progression. Moreover, a major study on this topic was not integrated: Mosler T, Conte F, Longo GMC, Mikicic I, Kreim N, Möckel MM, Petrosino G, Flach J, Barau J, Luke B, Roukos V, Beli P. R-loop proximity proteomics identifies a role of DDX41 in transcription-associated genomic instability. Nat Commun. 2021 Dec 16;12(1):7314. doi: 10.1038/s41467-021-27530-y. PMID: 34916496; PMCID: PMC8677849.

·      One important issue in family with pathogenic/likely pathogenic germline DDX41 mutations in the predisposition to myeloid malignancies but also to lymphoid malignancies, including follicular lymphoma and Hodgkin lymphoma. This should be integrated in the review.

·      Regarding specific therapies for MDS/AML patients having DDX41 germline mutation, authors strongly suggest the use of Lenalidomide as single agent. This is based on retrospective studies including less than 10 patients with DDX41 germline mutations, and one prospective case report about one patient (ref 54, Abou Dalle). The rational is based on location of DDX41 on the long arm of chromosome 5. But DDX41 is not located on the minimal deletion region implicated in response to Lenalidomide of MDS patients with 5q deletion. In my opinion, this is not sufficient to prone this therapy in patient who can benefit from others therapies like hypomethylating agents or intensive chemotherapy, also reported to have very good outcomes in DDX41 germline mutated patients.

Author Response

Dear reviewer,

Thank you very much for your kind input on our manuscript. We found your thorough review very helpful to improve our manuscript. Thanks to your feedback, we were able to update our manuscript with more updated information.

  • Two recent major publication in Blood on DDX41 in myeloid malignancies are not included in the review and should be integrated:

      - Duployez N, Largeaud L, Duchmann M, Kim R, Rieunier J, Lambert J, Bidet A, Larcher L, Lemoine J, Delhommeau F, Hirsch P, Fenwarth L, Kosmider O, Decroocq J, Bouvier A, Le Bris Y, Ochmann M, Santagostino A, Adès L, Fenaux P, Thomas X, Micol JB, Gardin C, Itzykson R, Soulier J, Clappier E, Recher C, Preudhomme C, Pigneux A, Dombret H, Delabesse E, Sébert M. Prognostic impact of DDX41 germline mutations in intensively treated acute myeloid leukemia patients: an ALFA-FILO study. Blood. 2022 Aug 18;140(7):756-768. doi: 10.1182/blood.2021015328. PMID: 35443031.

      - Li P, Brown S, Williams M, White T, Xie W, Cui W, Peker D, Lei L, Kunder CA, Wang HY, Murray SS, Vagher J, Kovacsovics T, Patel JL. The genetic landscape of germline DDX41 variants predisposing to myeloid neoplasms. Blood. 2022 Aug 18;140(7):716-755. doi: 10.1182/blood.2021015135. PMID: 35671390.

  1. We included Duployez et al. and Li et al. in our study, and updated table. Thank you very much for giving us an opportunity to learn and improve the manuscript.

  • Regarding age at onset, authors stated in the simple summary “younger age at diagnosis than the general AML/MDS population” and in the text “For MDS/AML patients with DDX41 mutations, the median age at diagnosis is 60 years (range, 57–70 years), which is slightly lower than that for patients with general MDS/AML”. This is not what is reported in the cited studies. Sébert et al. (ref 33) reported a median age of 69 years, Quesada et al. (ref 34) 70 years and Polprasert et al. (ref 35) 64 years. This was confirmed in the two recent studies (see major point 1). In my knowledge, MDS/AML patients with germline DDX41mutations are diagnosed at similar age than the sporadic patients DDX41 while type.
  1. We appreciated your input about the age at onset, and agreed that recent studies using larger data reported older age at onset. We updated the manuscript as below, following your input in the first paragraph of section “4. Clinical Presentation and Outcomes”

The age at diagnosis of MDS/AML patients with germline DDX41 mutations has been reported with a wide range from 57 to 70 years, but most studies reported that it is comparable to patients with general MDS/AML 34-36, 50, 58.

  • Regarding penetrance, the overall disease penetrance is currently not known as stated in recent review in the field: Klco et al, Nat review cancer (PMID: 33328584). We could speculate a low one, but no relevant study demonstrated it. In this way, authors should not claim that the penetrance is low.

  1. Thank you for pointing out the statement of penetrance. Following your comments, we updated our manuscript in section “3. Pathologic Features of Myeloid Neoplasms with DDX41 mutations” and section “7. “

Germline DDX41 mutations are passed from parent to child through autosomal dominant inheritance, but the penetrance of DDX41 mutations remains unclear. Only about 30% of patients with germline DDX41 mutations have a family history of hematologic malignancy, but a familial series study showed high penetrant patterns in some families.34, 48 Interestingly, DDX41-mutant MDS/AML has a male predominance, with a male-to-female ratio of 3:1 34, 50. The etiology of male predominance remains unknown. (Section 3)

However, the risk of malignancy and penetrance of disease in germline carriers remains unclear, and methods for detecting or preventing myeloid neoplasms in these individuals have not been established 66. (Section 7)

  • Regarding biological functions of DDX41 protein. The authors resumed in the abstract and the simple summary as “RNA metabolisms and innate immunity”. In my opinion, the overall function of this protein in the context of haematopoietic cells and how these mutations lead to haematopoietic neoplasms is also related to RNA splicing and processing, ribosomal biogenesis and cycle progression. Moreover, a major study on this topic was not integrated: Mosler T, Conte F, Longo GMC, Mikicic I, Kreim N, Möckel MM, Petrosino G, Flach J, Barau J, Luke B, Roukos V, Beli P. R-loop proximity proteomics identifies a role of DDX41 in transcription-associated genomic instability. Nat Commun. 2021 Dec 16;12(1):7314. doi: 10.1038/s41467-021-27530-y. PMID: 34916496; PMCID: PMC8677849.

  1. We appreciate giving us insight about updates in the field, about mechanism studies. We updated simple summary part as “So far mechanism studies revealed the mutations, which can be affected by both germline and somatic mutations, can be pathogenic by impairments in the normal function of genes involving RNA splicing and processing, ribosomal biogenesis, metabolism, cycle progression and innate immunity”. We included Mosler et al. in 4th paragraph of section 2. “DDX41 mutations and their role in MDS/AML Pathogenesis” as below.

Additionally, Mosler et al. proposed that increased level of R-loop, RNA-DNA hybrids and displaced strand of DNA, can be caused by dysregulation from loss of DDX41 in a recent study using quantitative mass spectrometry-based proteomics using human cells43. The study showed pathogenic variants of DDX41 mutations resulted in accumulation of double-strand break in human hematopoietic stem cells causing inflammatory responses. These enhanced inflammatory responses from R-loop accumulation and dysregulation of hematopoietic stem cell production were also shown in a study using zebra fish by Weinreb et al.44 These studies propose the role of DDX41 in genomic stability against R-loop generations.

  • One important issue in family with pathogenic/likely pathogenic germline DDX41 mutations in the predisposition to myeloid malignancies but also to lymphoid malignancies, including follicular lymphoma and Hodgkin lymphoma. This should be integrated in the review.

  1. Follicular lymphoma and Hodgkin lymphoma was mentioned as associated cancers in earlier section (last paragraph of section 2). In the section about family screening(section 7), we updated as below.

However, the risk of malignancy and penetrance of disease in germline carriers remains unclear, and methods for detecting or preventing myeloid neoplasms and other cancers in these individuals have not been established 66.

  • Regarding specific therapies for MDS/AML patients having DDX41germline mutation, authors strongly suggest the use of Lenalidomide as single agent. This is based on retrospective studies including less than 10 patients with DDX41 germline mutations, and one prospective case report about one patient (ref 54, Abou Dalle). The rational is based on location of DDX41 on the long arm of chromosome 5. But DDX41 is not located on the minimal deletion region implicated in response to Lenalidomide of MDS patients with 5q deletion. In my opinion, this is not sufficient to prone this therapy in patient who can benefit from others therapies like hypomethylating agents or intensive chemotherapy, also reported to have very good outcomes in DDX41 germline mutated patients.

  1. We appreciate your insight about treatment in DDX41 mutant AML/MDS patients. Following your input, we changed the treatment section (section 5) from “Treatment – Lenalidomide” to “Treatment – General approaches” and re-wrote the first 2 paragraphs as below.

  So far, there are no randomized studies focusing on patients with DDX41 mutations for specific therapeutic approaches. The treatment approaches of DDX41-mutant myeloid neoplasms generally follow standard care for general MDS/AML treatments. 2 recent studies reported treatment responses in DDX41 mutant patients. Li et al. reported a higher response rate at 78% and superior survival (not reached) in a matched case-control study including 28 patients with AML having DDX41 germline mutations. In the study, most (about 80%) of patients were treated with hypomethylating agents with or without venetoclax.59 Duployez et al., as mentioned above, also reported better response to intensive chemotherapy in patients with intermediate or adverse DDX41 germline-mutant patients in AML.

  Interestingly, a few studies showed a good response to Lenalidomide in MDS patients with DDX41 mutations36. Lenalidomide is a well-established standard therapy for the low-risk MDS with a 5q deletion60 Lenalidomide has demonstrated its efficacy in MDS/AML patients with DDX41 mutations, even in the absence of a full 5q deletion 36. Only 26% of patients with MDS with 5q deletion have the deletion at 5q35, where DDX41 is located at 36.

We sincerely appreciate your time and great review of our manuscript. We hope our updated manuscript is acceptable for publication in Cancers.

Thank you very much.

Sincerely,

Kunhwa Kim, MD MPH

Koji Sasaki, MD PhD

Reviewer 3 Report

Overall, this review is quite comprehensive and covers most major DDX41 publications. However, there are numerous typographical errors and variant annotation differences throughout the manuscript and table 1. The Simple Summary is not particularly well written and has a number of claims that are not correct when looking at all the data.

Information and data from the latest DDX41 reviews should be included (PMID: 35671390, 35443031). It is acknowledged that these came out only one day before this review was submitted, but there is significant data that is of importance for a review such as this.

Points to address:

1.)          Simple Summary - “one of the earliest studied genes” - there are numerous earlier ones before 2015 - RUNX1, CEBPA, GATA2, TERT, TERT. “Low penetrance” - the penetrance is not clear at this stage, as there are many reported cases of myeloid malignancy over 80 years of age and in many families studied, most individuals haven’t yet reach ages of this lateness of onset. “Younger age of diagnosis” – I do not think this is true when looking at all of the data including some of the newer publications).

2.)          On p2, line 56, it is stated “well describe pathogenesis” and “well-identified pathophysiology” (in Conclusion) – while several studies have nicely demonstrated processes that are impacted by germline variants in DDX41, the actual mechanism of pathogenesis that causes myeloid malignancy is still speculation – it has not be shown definitively.

3.)          It would be good to mention that the reason for the “male predominance” is unknown.

4.)          “The secondary somatic mutations resulting in the biallelic DDX41 mutations are believed to be similar to somatic mutations in the CCAAT enhancer binding protein alpha gene, CEBPA” – apart from being biallelic, are there any other similarities – the other types of somatic mutations are very different and CEBPA leads only to AML, not MDS.

5.)          “Notably, germline and somatic DDX41 mutations tend to have different locations on the gene.” – this is true for European variants, but not for some of the Asian groups where A500fs* co-occurs with somatic R525H. Of more interest is that it is quite unusual to have germline and somatic complete LOF variants co-occuring.

6.)          “p.M1I and p.M155I are predicted to generate small DDX41 isoforms that are potentially functionally defective 45, 47” – the M1I may generate predominantly the smaller isoform, the function of which has not been determined other than it lacks a NLS. M155I does not alter the amount of long or small isoform, and increasing evidence suggests it is a VUS and may not be pathogenic.

7.)          Table 1: clarify “Number of patients (%)” in the headings. Also do the last 2 columns, count patients and families together – clarify? p.R525H* - remove asterix. Be consistent – use either “X” or “*” but not both. For the Alkhateeb variants, p.Q41, p.R311 and p. Q368 should all have “*” added. Check all other variants for errors.

8.)          “…….DDX41 mutations are very rare among patients without cancer” – this is unclear – do you mean patients or carriers, and what type(s) of cancers?

9.)          “Because germline DDX41 mutations have low to moderate penetrance, the family members identified as germline DDX41 should determine the timing and methods of their surveillance.” – it is not clear how families can determine this…..informed or guided by hematologists???

10.)        “Other considerations for the screening and surveillance of patients and their families should include the prevention of secondary hematologic malignancies or autoimmune diseases in MDS/AML patients who have had disease remission and the prevention of primary hematologic malignancies or autoimmune diseases in germline mutation carriers.” – I don’t think screening or surveillance can “prevent” hematologic malignancies or autoimmune diseases – do you mean “early detection”?

11.)        “tailored approaches” – should this be tailored treatments or management?

12.)        Reword the last sentence to clarify.

Author Response

Dear reviewer,

Thank you for your kind input. Your review was very helpful to improve our manuscript significantly. We made changes in our manuscript addressing your points.

1.)          Simple Summary - “one of the earliest studied genes” - there are numerous earlier ones before 2015 - RUNX1, CEBPA, GATA2, TERT, TERT. “Low penetrance” - the penetrance is not clear at this stage, as there are many reported cases of myeloid malignancy over 80 years of age and in many families studied, most individuals haven’t yet reach ages of this lateness of onset. “Younger age of diagnosis” – I do not think this is true when looking at all of the data including some of the newer publications).

Thank you for sharing your insight. We agreed, and deleted all the parts regarding “earliest studied genes”, “low penetrance” and “younger age of diagnosis” in simple summary and entire manuscript.

2.)          On p2, line 56, it is stated “well describe pathogenesis” and “well-identified pathophysiology” (in Conclusion) – while several studies have nicely demonstrated processes that are impacted by germline variants in DDX41, the actual mechanism of pathogenesis that causes myeloid malignancy is still speculation – it has not be shown definitively.

Following your comments, we changed the statement to “Many studies have come out in past years, which provides better understanding in pathogenesis and clinical implications of DDX41 mutations.” Instead of “well-described pathogenesis”

3.)          It would be good to mention that the reason for the “male predominance” is unknown.

We updated section 3. stating “DDX41-mutant MDS/AML has a male predominance, with a male-to-female ratio of 3:1 34, 50. The etiology of male predominance remains unknown.”

4.)          “The secondary somatic mutations resulting in the biallelic DDX41 mutations are believed to be similar to somatic mutations in the CCAAT enhancer binding protein alpha gene, CEBPA” – apart from being biallelic, are there any other similarities – the other types of somatic mutations are very different and CEBPA leads only to AML, not MDS.

Following your comments, we updated the statement to “The pattern of biallelic mutations from secondary somatic mutations in DDX41 mutations also can be seen in somatic mutations in the CCAAT enhancer binding protein alpha gene, CEBPA in AML36.”

5.)          “Notably, germline and somatic DDX41 mutations tend to have different locations on the gene.” – this is true for European variants, but not for some of the Asian groups where A500fs* co-occurs with somatic R525H. Of more interest is that it is quite unusual to have germline and somatic complete LOF variants co-occuring.

Following your input, we changed paragraph as

“Ethnicity-based differences in the frequencies of DDX41 mutations have been noted. Several studies from cohorts of Asian patients showed distinct mutations profile. In a study of 28 Korean patients, the most common germline mutation was p.V152G, which was detected in 10 patients, followed by p.Y259C, p.A500fs, and p.E7* 45. Similarly, in a Japanese population, p.A500fs and p.E7X were the most frequent germline mutations, and p.R525H was the most common somatic mutation 53. In a study of Chinese patients, the most common mutations were c.935+4A>T and p.T360Ifs*33, which were detected in 3 patients each 54. This ethnic difference in mutations was shown in recent study by Li et al. with large data including 176 patients with germline mutations.50 In the study, p.A500fs was only identified in Asian patients, and 92% of patients with Y295C(n=12) was Asian patients. Majority, over 90%, of patients with p.M1I or p.D140fs were Caucasians. Interestingly, the study also showed more frequent loss of function in germline mutations (85% vs. 51%) in Caucasians than Asians.”

6.)          “p.M1I and p.M155I are predicted to generate small DDX41 isoforms that are potentially functionally defective 45, 47” – the M1I may generate predominantly the smaller isoform, the function of which has not been determined other than it lacks a NLS. M155I does not alter the amount of long or small isoform, and increasing evidence suggests it is a VUS and may not be pathogenic.

We agreed your comments, and we deleted the sentence. Thank you.

7.)          Table 1: clarify “Number of patients (%)” in the headings. Also do the last 2 columns, count patients and families together – clarify? p.R525H* - remove asterix. Be consistent – use either “X” or “*” but not both. For the Alkhateeb variants, p.Q41, p.R311 and p. Q368 should all have “*” added. Check all other variants for errors.

We changed the heading to “Number of patients or families (% of studied patients or families)”. We counted number of patients only in last 2 columns, so deleted the word families in headings. We tried to address the variants to look consistent in the table.

8.)          “…….DDX41 mutations are very rare among patients without cancer” – this is unclear – do you mean patients or carriers, and what type(s) of cancers?

We updated the sentence as “Family screening might be considered only for patients who have germline mutations because somatic DDX41 mutations are not heritable, and DDX41 mutations are very rare among general population 52

9.)          “Because germline DDX41 mutations have low to moderate penetrance, the family members identified as germline DDX41 should determine the timing and methods of their surveillance.” – it is not clear how families can determine this…..informed or guided by hematologists???

We updated the sentence as “As a first step, the timing and methods of surveillance for the family members identified as germline DDX41 needs to be established.”. Thank you.

10.)        “Other considerations for the screening and surveillance of patients and their families should include the prevention of secondary hematologic malignancies or autoimmune diseases in MDS/AML patients who have had disease remission and the prevention of primary hematologic malignancies or autoimmune diseases in germline mutation carriers.” – I don’t think screening or surveillance can “prevent” hematologic malignancies or autoimmune diseases – do you mean “early detection”?

Following your comments, we updated as below,

Other considerations for the screening and surveillance of patients and their families should include the early detection of secondary hematologic malignancies both myeloid and lymphoid malignancies, including follicular lymphoma and Hodgkin’s lymphoma, or autoimmune diseases in MDS/AML patients who have had disease remission and the early detection of primary hematologic malignancies or autoimmune diseases in germline mutation carriers.

11.)        “tailored approaches” – should this be tailored treatments or management?

We agreed that the words read vague. We changed the entire sentence as below,

“Further understanding of the clinical impact of the mutations including learning further prognostic or clinical information of DDX41 mutant AML/MDS patients, and studying if DDX41 mutations can be targetable for therapeutic or preventive applications would be essential.”

12.)        Reword the last sentence to clarify.

We updated the last sentence as below,

“Additionally, establishing the standardized approaches for surveillance/screening in the family members, germline-carriers would be interesting to follow.”

We also included and updated studies from Li et al. and Duployez et al. that you mentioned. Thank you very much.

We sincerely appreciate your sharing your insights and expertise with us. Thank you for giving us opportunity to improve the manuscript.

Kunhwa Kim, MD MPH

Koji Sasaki, MD PhD

Round 2

Reviewer 3 Report

This manuscript is now much better and easier to read and comprehend. All comments are tracked in the attachment.

Author Response

The manuscript has been revised accordingly. The revised version is attached.
